# Memristive FG–PVA Structures Fabricated with the Use of High Energy Xe Ion Irradiation

**DOI:** 10.3390/ma15062085

**Published:** 2022-03-11

**Authors:** Artem I. Ivanov, Irina V. Antonova, Nadezhda A. Nebogatikova, Andrzej Olejniczak

**Affiliations:** 1Rzhanov Institute of Semiconductor Physics, Siberian Branch of the Russian Academy of Sciences, 630090 Novosibirsk, Russia; nadonebo@isp.nsc.ru; 2Novosibirsk State Technical University, 630073 Novosibirsk, Russia; 3Faculty of Chemistry, Nicolaus Copernicus University, 87-100 Torun, Poland; aolejnic@chem.uni.torun.pl; 4Flerov Laboratory of Nuclear Reactions, Joint Institute for Nuclear Research, 141980 Dubna, Russia

**Keywords:** fluorinated graphene, high energy ion irradiation, graphene quantum dots, flexible memristors, FG–PVA active layer, pulse measurements, switching parameters

## Abstract

A new approach based on the irradiation by heavy high energy ions (Xe ions with 26 and 167 MeV) was used for the creation of graphene quantum dots in the fluorinated matrix and the formation of the memristors in double-layer structures consisting of fluorinated graphene (FG) on polyvinyl alcohol (PVA). As a result, memristive switchings with an ON/OFF current relation ~2–4 orders of magnitude were observed in 2D printed crossbar structures with the active layer consisting of dielectric FG films on PVA after ion irradiation. All used ion energies and fluences (3 × 10^10^ and 3 × 10^11^ cm^−2^) led to the appearance of memristive switchings. Pockets with 10^3^ pulses through each sample were passed for testing, and any changes in the ON/OFF current ratio were not observed. Pulse measurements allowed us to determine the time of crossbar structures opening of about 30–40 ns for the opening voltage of 2.5 V. Thus, the graphene quantum dots created in the fluorinated matrix by the high energy ions are a perspective approach for the development of flexible memristors and signal processing.

## 1. Introduction

In the Internet-of-things (IoT) era, energy-efficient and data processing speeds become the bottlenecks for further progress in informative operation technologies. High-density memories with the analog switching type are the basic need for neuromorphic computing [1,2]. Various analog-programmable emerging nonvolatile memory devices and, first of all, resistive random-access memories (RRAM), are cutting-edge technologies [3,4]. Because of the simple two-terminal design, a low-cost, cost-efficient, ultrahigh density, high-speed operating, etc., RRAM has become the most prospective candidate for emerging nonvolatile memory devices [5,6,7,8].

RRAM, also known as memristor, is an electronic device based on a metal-insulator-metal (MIM) structure, in which the internal resistance state is the recorded history of the applied voltage (or current), and the switching is the change of the state from a high-resistance state to a low-resistance state and back. The extremal parameters of RRAM are known for HfO_2_ (extremely small device sized ~2 nm and only the minimal energy with a 6-bit/cell storage capacity) [9,10], and high-speed resistive switching (~few nanoseconds) is obtained for memristors based on AlN films [11].

As the RRAM memristor size is scaled down to nanometer ranges, controlling the size and shape of the nanostructure electrode is very critical as RRAM defects can play an important role as a way to improve the electric field localization, increase the charge trapping ability, and also it can serve different purposes based on the structure design [2].

A stable bipolar resistive switching effect with the ON/OFF current ratio amounting from one to 4–5 orders of magnitude is found for two-layer films of partially fluorinated graphene with graphene quantum dots (GQDs) and polyvinyl alcohol [12]. Resistance switches were observed when the fluorination degree was relatively low (F/C ~20–25%), and quantum dots of graphene are the basis for the FG film conductivity. These structures are perspectives for nonvolatile memory cells, for the information storage, and data processing of flexible and printed electronics. The main problem of these structures is the size and quality of GQDs in the FG layer. GQDs are formed during the fluorination process in the hydrofluoric acid solution [13,14,15]. To stop fluorination at a particular stage and create partially fluorinated material with required GQDs is a complicated task due to the strong fluorinated process dependence on a lot of graphene parameters (structural, morphological, and electrical).

The new approach for creating GQDs in the fluorinated matrix, which is necessary for the memristive switchings, is suggested with the use of irradiation by heavy high-energy ions. According to our results, irradiation leads to the FG defluorination in the ion track area with the formation of 3–5 nm GQDs [16]. In the present study, Xe ion irradiation was used to create the GQDs in 2D printed crossbar structures with an active dielectric FG film layer on PVA. As a result, the memristive effect with the ON/OFF current relation of ~2–4 orders of magnitude was observed in all irradiated structures, and the opening time was 30–40 ns. Thus, this approach is promising for the formation of two-phase memristor materials.

## 2. Materials and Methods

### 2.1. FG Suspension Preparation

The first stage of sample preparation is the fabrication of graphene suspensions. In this study, liquid exfoliation of natural graphite in combination with ultrasonic treatment and centrifugation was used. The fluorination process proceeds in a weak (~3–7%) aqueous hydrofluoric acid solution [13,14,15]. In the process of fluorination, additional exfoliation greatly reduces the thickness of fluorinated graphene flakes to several monolayers. Simultaneously, the films obtained from the suspensions become smoother and more uniform [13]. As a result, we created the FG suspension with a fluorination degree of F/C ~30–35%. This fluorination degree is the maximum of our fluorination approach and allows us to create dielectric films.

### 2.2. Fabrication of Memristor Structures

PVA was deposited on a silicon substrate or PET (polyethylene terephthalate) using a spin processor, and the deposited PVA layer thickness was 30 nm. In the present study, we have used PVA with a low molecular weight of 13,000–23,000 (Sigma-Aldrich, St. Louis, MI, USA). Fluorinated graphene has a fluorinated degree ~30–35%. The thickness of printed FG layers on PVA was varied from 4 nm to 20 nm with the step of 4 nm (5, 10, 15, 20, and 25 printed layers). The SEM and optical image of FG–PVA film, the schematic image of crossbar memristor structures, and the array of memristors printed on PET are presented in Figure 1. The active memristor layer size in the structures is 900 × 900 μm^2^. Ag inks were used for the bottom and top contacts with the line width of 300 × 300 μm^2^ and the thickness of 200 nm. The upper Ag electrode for the crossbar structures was created after the ion irradiation of the active layers. The Ag contacts sheet resistivity was ~20 Ohm/sq. In some cases, silicon substrates were used as bottom contacts.

The polyvinyl alcohol used in the present study is short chains, and for this reason, PVA is a dielectric material. Fluorinated graphene before irradiation is also a dielectric material. The current-voltage characteristics for structures with an active layer created from PVA only or structures with FG layer are given in the Appendix A.

### 2.3. Ion Irradiation Regimes

Irradiation was carried out by the Xe ions with the energy of either 26 MeV or 167 MeV; the fluences were 3 × 10^10^ and 3 × 10^11^ cm^−^^2^. The ion flux in the ion beam was 5.7 × 10^8^ ions/cm^2^s^1^ at room temperature, in a vacuum at the pressure of 6.3 × 10^−^^6^ Torr. The ion projected range (the ion penetration depth into the Si substrates) was equal to ~5 and ~19 μm for 26 and 167 MeV, respectively. Thus, the main part of defects introduced by ions are located in the substrates, and, in the thin active layer (35–50 nm, which consist of 30 nm of PVA and 5–20 nm of FG) of memristor structures, ion losses were connected with the ion interaction with the electron subsystem of the structures and atom ionization (electron losses). The ion irradiations were carried out on an ion-beam line for the applied research on the IC-100 cyclotron at FLNR JINR, Dubna.

### 2.4. Experimental Techniques

The structural and electric properties of the films were studied by means of atomic force microscopy (AFM), scanning electron microscopy (SEM), and the experimental complex for testing electric properties. The film and structure morphologies were examined on Hitachi SU8220 scanning electron microscopes at electron–beam energies ranging from 2 to 15 keV. A Solver PRO NT-MDT scanning microscope was used for obtaining the AFM images of substrate and sample surfaces and for measuring the film thicknesses. The measurements were carried out in contact and semi-contact modes. The AFM probe tip rounding radius was ~10 nm. 

To visualize fluorinated graphene in the active layer, such AFM measurement modes as MAG and MAGSin were used. The signal associated with the probe phase is proportional to the product of the alternating signal amplitude at the modulation frequency (MAG is a signal proportional to the amplitude of cantilever oscillations) multiplied on the sinus of the cantilever oscillations phase shift relative to the reference signal. Accordingly, the phase shift value of the cantilever oscillations relative to the reference signal consists of the sum of the excitation signal phase shift relative to the reference and the cantilever oscillations phase shift relative to the excitation signal accurate to a certain “hardware” constant. The oscillation phase is more sensitive, in comparison with the amplitude, to changes in the interaction of the probe and the surface, and in particular to changes with local differences in the surface adhesion and viscoelasticity.

The film sheet resistance was studied with the use of four-probe JANDEL equipment and an HM21 Test Unit. For measuring current-voltage (I–V) characteristics, a Keithley picoamperemeter (model 6485) was used. Pulsed measurements were recorded on oscilloscope Keysight DSOX3012T.

The films and the structures were fabricated by the 2D printing method. A Dimatix FUJIFILM DMP-2831 printer equipped with a DMC-11610 printing head with 16 nozzle carriers of about 20-μm in diameter was used for printing. The printing process was implemented on both solid and flexible substrates. As flexible substrates, polyethylene terephthalate (PET) substrates with an adhesive coating (Lamond) and thickness of ~200-μm were used. During the printing process, the substrate temperature was maintained at 60 °C.

## 3. Results

### 3.1. Memristor Active Layer Images and Parameters

The SEM images of the two crossbar memristor structure variants with the same FG–PVA active layer (Ag–FG–PVA–Si and Ag–FG–PVA–Ag) and the sketches of the crossbar memristor structures are given in Figure 1. The active layer is transparent, and, in the second case, it is colored in Figure 1b. The active FG–PVA layers, on a larger scale, are given in Figure 2a,b. One can see the FG flakes in both cases.

The sketch image of the active layer is presented in Figure 2c. As shown in Figure 3, the ion irradiation leads to the formation of thermally expanded flakes with a sized ~100 nm and the cracking of the FG flakes on small nanoparticles. The higher the ion energy, the higher the number of the smaller nanoparts. The appearance of small (~3 nm) graphene quantum dots embedded in FG nanoparticles is also expected in the irradiated films, according to our previous study [16]. It is well known that the ions deeply penetrate the substrate, and, partially, their energy is lost to modify the near-surface FG layers. We assume that graphene areas with a diameter of 1–3 nm could be created in separate tracks of fast heavy ions in the dielectric FG flakes. Rapid local heating leads to a partial defluorination of nano areas with subsequent structural changes and the formation of nanometer-sized graphene islands. These GQDs are observed by HREM (see Appendix A). The local shock heating in ion tracks is suggested to be the leading force driving the defluorination process. Dielectric FG films with small quantum dots may offer prospects to correct the functional properties of printed FG-based crossbar memristors. For Ag–FG–PVA–Ag structures, the active layer is seen as a suspended film in Figure 2d. 

In the case of irradiation with 26 MeV Xe ions, FGs particles broke into small ones with lateral dimensions 30–50 nm and the thickness of 10–20 nm. After the irradiation with 167 MeV ions, the particles are still gathered in clusters 100–150 nm with a thickness of 15–25 nm. Initial graphene flakes had a lateral size of 150–300 nm and a thickness of 0.34–5 nm.

The AFM images of the pristine and irradiated active layers FG–PVA recorded in different regimes (height, MagSin, and Mag) are demonstrated in Figure 4. The use of varying record regimes allows us to observe FG and PVA separately in the active layer. FG is seen as a black part of the surface in the MagSin images and a smooth part in the Mag images. One can see that both FG and PVA have to take part in the conductivity between the contacts in crossbar structures. The considerable relief of the FG–PVA layer is created during the FG layer application on water-soluble PVA. The comparison of AFM images for the pristine active layer and the irradiated ones demonstrates a decrease not only in the FG nanoparticle size but also in the size of surface morphology elements.

The current-voltage characteristics for the Ag–FG–PVA–Si structures before and after irradiation are given in Figure 5. We like to stress that the pristine crossbar structures demonstrate low currents (10^−^^11^–10^−^^9^ A) due to the high fluorination degree of the top FG layer and PVA used there have practically no resistive switchings. The used PVA is the dielectric layer. 

After the Xe ions irradiation (energies 26 and 167 MeV and fluences 3 × 10^10^ and 3 × 10^11^ cm^−^^2^), the resistive switchings were observed in all structures. The ON/OFF current ratios, practically in all cases, were 2–4 orders of magnitude regardless of the FG layer thickness. The current values in the crossbar structure increased from 10^−^^11^–10^−^^9^ A to 10^−^^7^–10^−^^6^ A. It is worth mentioning that, after irradiation, the PVA film (without FG layer) does not show any conductivity or resistive switchings (see Appendix A). The current-voltage characteristics for the FG films with fluorination degree ~20–25% typically demonstrate resistive switchings with ON/OFF current ratio of about 1–2 orders of magnitude (see Ref. [14]).

The typical current-voltage characteristics for the printed Ag–FG–PVA–Ag crossbar structure measured after the irradiation by Xe ions with the energy of 26 MeV and the fluence of 3 × 10^11^ cm^−2^ are given in Figure 6a. Before irradiation, the current values were equal to 10^−^^9^–10^−^^8^ A. After irradiation, the currents were increased up to 10^−^^3^–10^−^^2^ A. The higher currents before and after irradiation are caused by Ag bottom contact with a large relief. The opening voltages in this type of structure are strongly decreased compared to Ag–FG–PVA–Si down to ~0.1–0.2 V. The current-voltage characteristics in coordinates ln(I) versus ln(V) for one of these structures are given in Figure 6b. These coordinates correspond to the space charge limited conduction (SCLC). Practically the same slope (1.0–1.3) of all curves means that the traps involved in the conduction do not change. 

### 3.2. Pulse Measurements

The results of the current switchings for one of the irradiated structures are demonstrated in Figure 7. The sketch of the tested voltage pulses is given in the inset of Figure 7b. More than 10^3^ pulses were passed through each sample. This test does not cause a change in I_ON_ and I_OFF_ currents. The Set-read-Reset-read complete cycle presents in the inset in Figure 7b.

The current in the open state as a function of the FG layer thickness and the ion energy for all irradiated structures are presented in Figure 8a,b. The maximum current was observed for the 10–12 nm FG thickness after the irradiation with the ion energy of 26 (167) MeV for both fluencies. It was found that the ion fluence does not practically affect the open-state current for the active layer thickness of 7–12 nm. A higher increase in the active layer thickness leads to a current increase. This fact is connected with the formation of GQDs predominantly near the surface (in the ~10 nm thick layer). 

The variation of the voltage pulse duration (Figure 8c,d) allows us to determine that the time of 30–40 ns is enough for the opening of crossbar structures. 

## 4. Discussion

Unexpected technological approaches often allow one to obtain promising solutions for the development of the new-generation devices in a wide spectrum of applications (for instance, [6,17,18]).

In recent years, much attention has been paid to the studies related to nanostructure irradiation with various ion beams. The interest in these studies is associated not only with obtaining fundamental knowledge about the processes of radiation damage and its evolution in nanostructured objects but also with the practical application of ionizing radiation for the possibility of directed modification of the properties of nanostructures. The theory of properties nanomaterial modification by ionizing radiation is based on the theory of a thermal peak arising from the interaction of an incident ion with a substance and the subsequent energy transfer to the electron and nuclear subsystems for very short periods, thereby causing a sharp temperature increase in the structure, comparable, in some cases, with the material melting point. The distinctive feature of ion radiation from a similar thermal effect is that the modification of the properties of nanostructures during irradiation occurs locally along the entire path of ions in the material, while, during the thermal annealing, changes occur throughout the material bulk [19].

In two-dimensional materials, the tuning of intrinsic material properties inevitably occurs through modifications on the atomic scale. Thus, 2D material science deals with defect engineering in the widest sense, that is, with the possibility of controlled removal, ddition, or manipulation of atoms, and as such, represents the key to unlocking the full potential for applications [20]. Ion irradiation, with the possibility of controlled introduction of energy ions, provides an alternative simple top-down route for graphene modifications and the formation of graphene quantum dots [21,22,23,24].

We assume that the formation of GQDs took place in the ion track areas. As mentioned above, when the ion passed through the film, a large amount of heat (up to 1500–3000 °C) was released locally in the few-nanometer size tracks for a short time (~10^−^^12^–10^−^^11^ s) [25]. This local heating leads to the defluorination process. The images of GQDs obtained with the use of HREM are given in (Appendix A). The FG film thickness was varied from 4 to 20 nm. We suggest the formation of a vertically ordered system of GQDs in tracks, which leads to weak dependence of currents in irradiated crossbar structures (see Figure 8a,b) for FG thickness range of 4–12 nm due to overcoming of the percolation threshold already at an ion dose of 3 × 10^10^ cm^−^^2^. For higher FG thickness, a decrease in current is most likely connected with the suppressing of fluorite removal when going deeper into the film.

According to the data presented above, the irradiation by Xe ions with an energy of 26–167 MeV and fluences of 3 × 10^10^ cm^−^^2^ and 3 × 10^11^ cm^−^^2^ changes the structure of active layers (formation of GQDs, fragmentation of the particles) and introduces some active centers, which leads to the observation of resistive switchings. The observation of resistive switchings after irradiation is the main finding of the present research. Based on our previous study [16], we can state that, due to defluorination, ion irradiation leads to the formation of graphene quantum dots in FG, and, most likely, electrically active centers in PVA or at the FG–PVA interface. This observation correlates with the known data in that chalcogenide atoms are easily removed from a 2D material (MoS_2_) upon irradiation due to the lower binding energy of chalcogenide atoms compared to transition metal atoms [20,26].

The mechanisms of the resistance-change phenomenon can be classed into two main types. The first one connects with the resistive switchings due to the formation/disintegration of conductive channels or filaments with different origins [5,9]. The second type of switching mechanism implies the participation of some electrically active centers (GQDs and traps) that can change their conductivity under applied voltage [12]. In the last case and in our materials, the open state current depends on the area of the structure due to variation in the number of conductive passes formed in structures. Thus, the resistive switching effect in our structures has described the conduction over quantum dots proceeding with the participation of active traps at the GQDs–PVA interface. GQDs provide the permanent conductive part of the pathways, whereas the traps at the GQDs–PVA interface are activated/deactivated only upon the applied voltage [12].

As is shown in Figure 3, the ion irradiation leads to the formation of thermally expanded flakes with a size of ~100 nm and cracking of the FG flakes on small particles with an ion energy increase. The appearance of small (~3 nm) graphene quantum dots embedded in FG nanoparticles is also expected in the irradiated films, according to our previous study [16]. After the Xe ions irradiation with both energies (26 and 167 MeV) and fluences (3 × 10^10^ and 3 × 10^11^ cm^−^^2^), the resistive switchings were observed in all irradiated structures. The ON/OFF current ratios practically in all cases were 2–4 orders of magnitude regardless of the FG layer thickness. The current values in the crossbar structures are increased from 10^−^^11^–10^−^^9^ A to 10^−^^7^–10^−^^6^ A. It is worth mentioning that, after irradiation, the PVA film (without FG layer) does not show any conductivity or resistive switchings. 

At the current-voltage characteristics in coordinates ln(I) versus ln(V) corresponding to the space charge limited conduction (SCLC) for crossbar structures, a linear fitting is observed. The same conductive mechanism was found for the case of Ag–FG–PVA–Ag crossbar structures with the FG–PVA active layer, in which GQDs are formed due to low fluorination [12]. SCLC is connected with the carrier tunneling between the traps. These traps are suggested to be related to GQDs in the FG matrix. The barrier height at the GQDs–FG interface was estimated in [17] as 0.48–0.50 eV for the similar FG layers. These values can be varied depending on the GQD size. For small GQDs (3–5 nm), the barrier height is strongly decreased. Practically the same slope (1.0–1.3) of all curves means that the traps involved in the conduction do not change.

In our study, the resistive switching mechanism is most likely associated with the activation with an applied voltage of the localized states at the FG–PVA interface, which, in turn, leads to the formation of paths for the electric current flow between the GQDs. Thus, on the one hand, PVA plays the role of a porous framework. On the other hand, PVA is involved in the surface states formation because it increases the ON/OFF current ratio. The strong current increase in the open state is associated with the percolation transition in the active layer, i.e., with the appearance of current paths after the voltage application.

One more interesting finding is connected with the fact that the current in the open state is very weak depending on the FG thickness in the studied range of fluences and energies. We have suggested that the GQDs formed in one ion track under irradiation have good electric connections or one GQD multilayer with a thickness up to ~10 nm is formed due to irradiation. Thus, in [27], the possibility of stitching individual graphene sheets into a common three-dimensional structure using ion irradiation was shown, and according to [28], the passage of heavy Au ions through the PMMA–graphene–Cu trilayer structure leads to creating rectilinear damage zones, the so-called latent tracks, all along the ion trajectory in the structure.

It was demonstrated that FG–PVA crossbar structures have good flexibility [12]. The currents in open and closed states are not practically changed up to the tensile strain due to the structure bending up to 6.5%. For a higher strain (6.5–8%), the ON/OFF ratio is decreased. Thus, the considered structures with the active FG–PVA layer are perspectives for the development of flexible memristors. The use of 2D printed technologies to create these memristors allows one to elaborate on the cheap variant of flexible memristor technologies.

## 5. Conclusions

A new approach for the creation of the memristive material based on the active FG–PVA layer using irradiation with high energy heavy ions was developed. Xe ions with energies 26 and 167 MeV and the fluences of 3 × 10^10^ cm^−^^2^ and 3 × 10^11^ cm^−^^2^ were used for irradiation. It was shown that irradiation causes the appearance of the stable effect of resistive switchings in crossbar structures with an active layer consisting of the fluorinated graphene film printed on PVA with a thickness of 35–50 nm. This fact confirms, once again, that the effect of high-energy ions in FG leads to partial defluorination of the material and the formation of graphene quantum dots, which, together with localized states at the FG–PVA interface, form the electric current flow paths. The ON/OFF ratio of currents, in this case, is ~2–4 orders of magnitude. The switching time is about 30–40 ns, and the endurance characteristics demonstrate the 2 × 10^3^ switching cycles without any changes in the ON/OFF current ratio. Thus, the ion irradiation used is a promising approach for the formation of the required active layer properties of FG-based flexible memristors.

## Figures and Tables

**Figure 1 materials-15-02085-f001:**
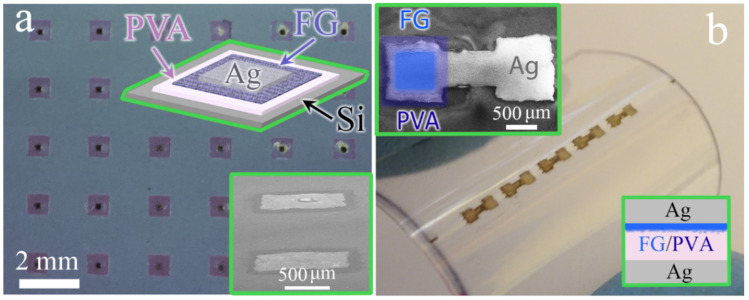
(**a**) SEM images of the two crossbar memristor structure variants and, in the insert, the sketch of the crossbar memristor structures. (**b**) Optical photo of the 2D printed crossbar memristor structures array on PET without top Ag contact to the active FG–PVA layer of one of the structures. The insert shows the SEM image of the one structure with a colored active FG–PVA layer. The active layer size is 900 × 900 μm, and the Ag contact size is 300 × 300 μm. The crossbar memristor structures are given in the inserts.

**Figure 2 materials-15-02085-f002:**
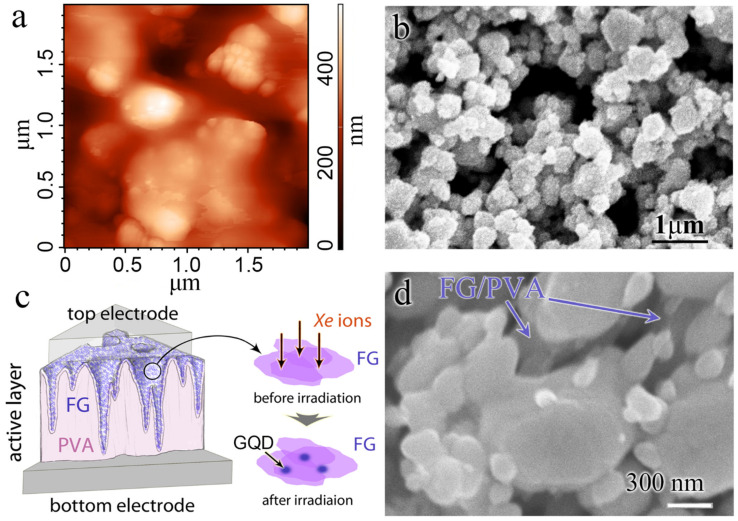
(**a**) AFM image of active FG–PVA layer of the crossbar structures on the Si substrate before irradiation. (**b**) SEM image of active FG–PVA layer of the crossbar structures on the Ag printed layer before irradiation (**c**) Sketch of the active FG–PVA layer in the crossbar structures and the formation of the GQD after high energy ion irradiation. (**d**) SEM image of active FG–PVA layer of the crossbar structures on the Ag printed layer after irradiation (167 MeV, 3 × 10^11^ cm^−2^).

**Figure 3 materials-15-02085-f003:**
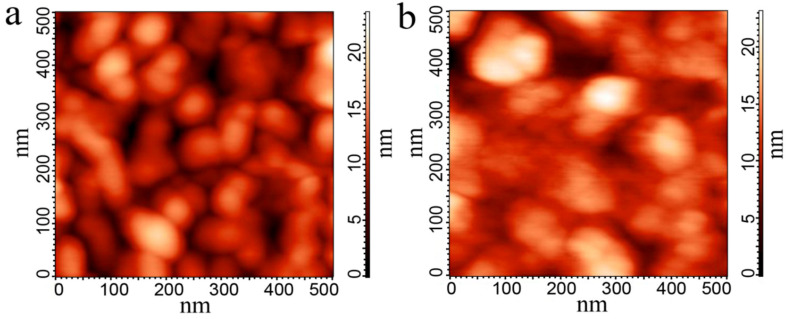
AFM images of the active layer of the Ag–FG–PVA–Si crossbar structures irradiated by Xe ions with the fluence of 3 × 10^11^ cm^−^^2^ and the energy of 26 MeV (**a**) and 167 MeV (**b**). The FG layer thickness is 7–10 (**a**) and 17–20 nm (**b**).

**Figure 4 materials-15-02085-f004:**
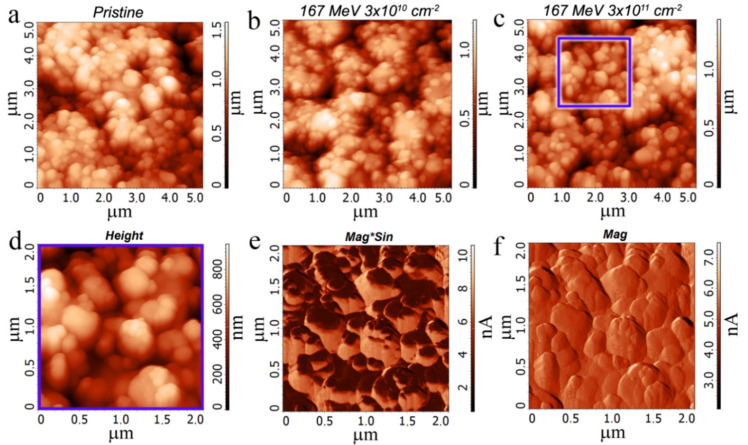
(**a**–**c**) AFM images of the pristine (**a**) and 167 MeV ion irradiated active FG–PVA layers. The ion fluence was equal to 3 × 10^10^ (**b**) and 3 × 10^11^ cm^−^^2^ (**c**). An enlarged surface fragment (**c**) is shown in (**d**). (**d**–**f**) The FG–PVA film irradiated with 167 MeV Xe ions (fluence was 3 × 10^11^ cm^−2^) was recorded in different regimes (height, MagSin, and Mag).

**Figure 5 materials-15-02085-f005:**
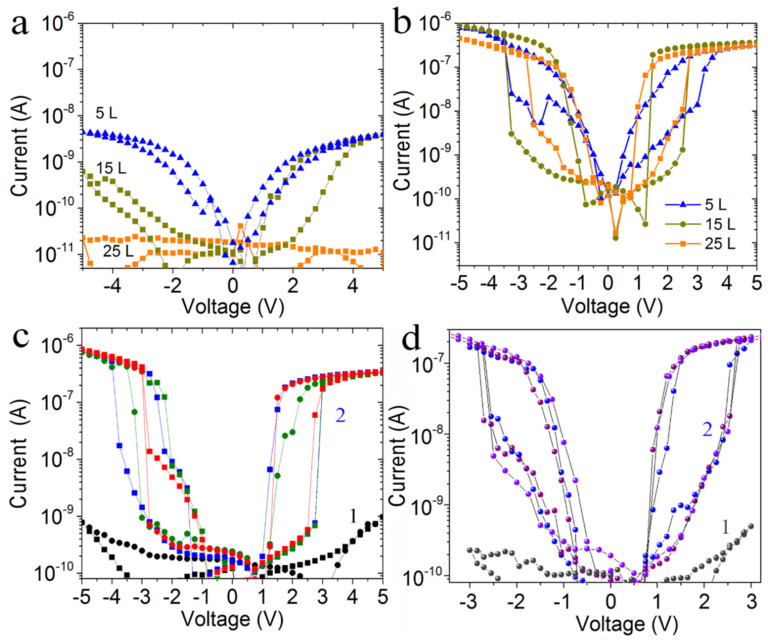
Current-voltage characteristics for the printed Ag–FG–PVA–Si crossbar structures with different thicknesses (5 L corresponds to 4 nm, 15 L to 13 nm, and 25 L to 20 nm) measured before and after the irradiation by Xe ions with the fluence of 3 × 10^11^ cm^−2^. (**a**,**b**) I–V curves before (**a**) and after (**b**) the irradiation by ions with energy 26 MeV and the fluence 3 × 10^11^ cm^−2^. (**c**,**d**) I–V curves measured before (1) and after (2) the irradiation: (**c**) FG thickness is 15 L (12 nm) and Xe ions with the energy of 26 MeV, (**d**) FG thickness is 25 L (20 nm) and Xe ions with the energy of 167 MeV. The contact area is 300 × 300 µm^2^.

**Figure 6 materials-15-02085-f006:**
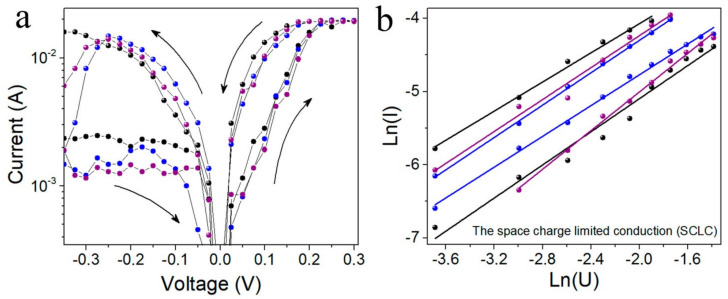
(**a**) Current-voltage characteristics for the printed Ag–FG–PVA–Ag crossbar structure measured after the irradiation by Xe ions with the energy of 167 MeV and the fluence 3 × 10^10^ cm^−2^. (**b**) The ON-state current in the coordinate of the space charge limited conduction (SCLC) ln(I) versus ln(V) for a few measurements given in (**a**).

**Figure 7 materials-15-02085-f007:**
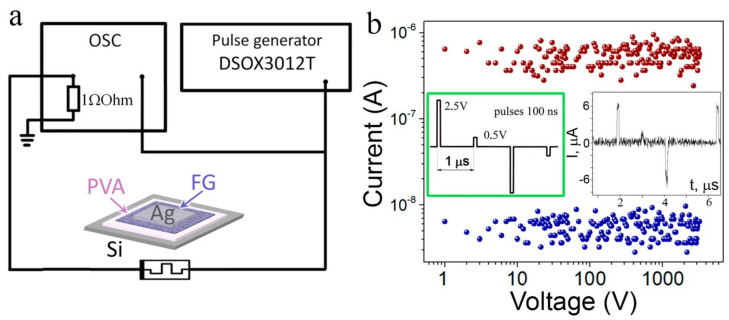
Current switchings for the structure irradiated with Xe ions (167 MeV, 3 × 10^11^ cm^−^^2^) with the FG thickness of 17–20 nm: (**a**) Circuit for measuring the pulse characteristics; (**b**) The I_ON_ and I_OFF_ currents as a function of pulse number. The sketches of the measured structures and the tested voltage pulses are given in the insets of (**a**) and (**b**), respectively. The second inset in (**b**) presents the Set-read-Reset-read cycle for the tested structure.

**Figure 8 materials-15-02085-f008:**
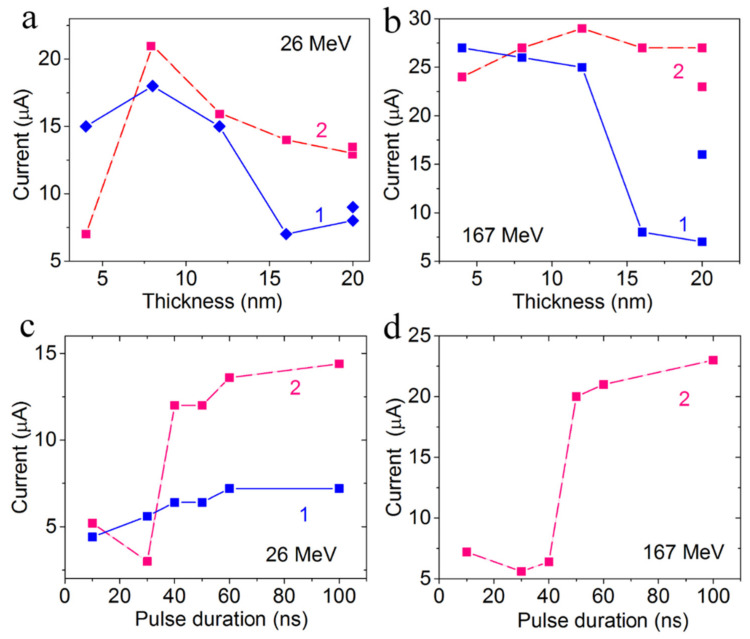
(**a**,**b**) Current in the memristive structures in the open state as a function of the FG layer thickness for irradiation with different Xe ion energies: (**c**,**d**) Open-state current versus opening pulse duration for the structures with the FG thickness of (**c**) 10–12 nm and (**d**) 17–20 nm. The irradiation regimes are as follows: the energies were (**a**,**c**) 26 MeV and (**b**,**d**) 167 MeV, the fluences were: 1–3 × 10^10^ cm^−^^2^, 2–3 × 10^11^ cm^−^^2^. The voltage pulse is 2.5 V.

## Data Availability

Not applicable.

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
