# Peer review of "Memristive FG–PVA Structures Fabricated with the Use of High Energy Xe Ion Irradiation"

_materials, 2022, doi:10.3390/ma15062085_

Round 1
Reviewer 1 Report
Conventional memristor-based research is greatly advanced by the findings presented in this publication. This study's findings are stunning, and they hold a great deal of promise. Prompt publication is recommended because of its relevance.
1) The idea of using Xe ion irradiation is interesting. However, the detailed underlying mechanism of forming graphene quantum dots is not discussed by authors.
2) To ensure a better understanding of the device performance for the readers, the switching speed of the hybrid memory device and the Set-read-Reset-read complete cycle under pulse mode should be supplemented.
3) What is the switching behavior without FG? The authors claimed that the presence of FG enhanced switching Characteristics. Please give a comprehensive supplementation on this.
4) 2D material recently emerged as an alternative to existing memristive material. In recent years there has been tremendous experimental improvement in memristor using 2D material (Nat Electron 4, 348–356 (2021); Small 2021, 17, 2006760) Authors may need to discuss those works in revised manuscript.
Author Response
First of all, we would like to thank you for your interest in our work and its high evaluation.
1) The idea of using Xe ion irradiation is interesting. However, the detailed underlying mechanism of forming graphene quantum dots is not discussed by authors.
In the Discussion, we have added a paragraph with some details of GQD formation (page 10, 3rd paragraph).
2) To ensure a better understanding of the device performance for the readers, the switching speed of the hybrid memory device and the Set-read-Reset-read complete cycle under pulse mode should be supplemented.
For a better understanding of the device performance in Fig.7b, we had added the insert, which demonstrates the Set-read-Reset-read complete cycle. The related words were now given in the text on page 8 before fig.8.
3) What is the switching behavior without FG? The authors claimed that the presence of FG enhanced switching Characteristics. Please give a comprehensive supplementation on this.
One more new paragraph was added in the Discussion (page 10, last paragraph). The mechanisms of resistive switchings are discussed in this paragraph. Please, see also Supplementary Fig.S1.
4) 2D material recently emerged as an alternative to existing memristive material. In recent years there has been tremendous experimental improvement in memristor using 2D material (Nat Electron 4, 348–356 (2021); Small 2021, 17, 2006760) Authors may need to discuss those works in revised manuscript.
Thank you for the interesting references. We have included them in the Introduction and Discussion.

Reviewer 2 Report
The authors developed a new approach based on the irradiation by heavy high energy ions to produce graphene quantum dots in the fluorinated matrix and the formation of the memristors in double-layer structures consisting of fluorinated graphene (FG) on polyvinyl alcohol (PVA). Interestingly, the 2D printed crossbar structures with the active layer consisting of dielectric FG films on PVA after ion irradiation showed memristive switchings with an ON/OFF current relation ~2-4 orders of magnitude. Additionally, pockets with 10^3 pulses were passed through each sample for their testing, whereas, the results of pulse measurements showed the time of crossbar structures opening of about 30-40 ns for the opening voltage of 2.5 V. The results suggest that the graphene quantum dots created in the fluorinated matrix by the high energy ions are a perspective approach for the development of flexible memristors and signal processing. The work is interesting and can be published in Materials if the following issues can be addressed:
1- The authors should cite the papers https://doi.org/10.1016/j.carbon.2020.10.045 and https://doi.org/10.1021/acsaelm.0c00838 in the introduction section for better reviewing the application of carbon-based materials for electronic devices.
2- More details about the materials used in the study need to be provided.
3- Why was the size of FG particles irradiated with 26 MeV Xe ions smaller than that of those irradiated with 167 MeV ions?
4- Figures 5-8 need to be improved. their legends are not consistent and unclear, especially for Figures with several data colors.
5- Why was the maximum current achieved by the FG thickness of 10-12 nm after the irradiation with the ion energy of 26 (167) MeV?
6- Why did the ion fluence not affect the open-state current for the active layer thickness of 7-12 nm?
Author Response
List of corrections and answers for the reviewer
First of all, we would like to thank you for your interest in our work and its high evaluation.
1- The authors should cite the papers https://doi.org/10.1016/j.carbon.2020.10.045 and https://doi.org/10.1021/acsaelm.0c00838 in the introduction section for better reviewing the application of carbon-based materials for electronic devices.
We have added more references in the Introduction and Discussion
2- More details about the materials used in the study need to be provided.
We have added some details about the used materials marked with yellow on pages 3, 4, 7.
3- Why was the size of FG particles irradiated with 26 MeV Xe ions smaller than that of those irradiated with 167 MeV ions?
Pristine FG particles are the same in both cases. Under irradiation, the particles are thermally expanded and then particles are separated on the nanoparts. The higher the ion energy, the higher number of the smaller nanoparts.
4- Figures 5-8 need to be improved. their legends are not consistent and unclear, especially for Figures with several data colors.
The captures of Figures 5-8 are corrected.
5- Why was the maximum current achieved by the FG thickness of 10-12 nm after the irradiation with the ion energy of 26 (167) MeV?
The FG film thickness was varied from 4 to 20 nm. We suggest the formation of a vertically ordered system of GQDs in tracks which leads to weak dependence of currents in irradiated crossbar structures (see Fig. 8a,b) for FG thickness range of 4-12 nm due to overcoming of the percolation threshold already at ion dose of 3x1010 cm-2. For higher FG thickness decrease in current most likely connected with suppressing of fluorite removal with going deeper into the film.
This explanation is added in the Discussion, page 10, 3rd paragraph.
6- Why did the ion fluence not affect the open-state current for the active layer thickness of 7-12 nm?
Fluence 3x1010 cm-2 is high enough for the formation of the high density of GQDs for overcoming the percolation threshold. See explanation added in the Discussion, page 10, 3rd paragraph.
